# Mate Selection on *Anastrepha curvicauda*: Effect of Weight, Age, and Virginity

**DOI:** 10.3390/insects14040317

**Published:** 2023-03-26

**Authors:** Patricia Villa-Ayala, Javier Hernández-Reynoso, Alfredo Jiménez-Pérez

**Affiliations:** Laboratorio de Ecología Química de Insectos, Centro de Desarrollo de Productos Bióticos, Instituto Politécnico Nacional, CEPROBI # 8, San Isidro, Yautepec 62731, Mexico

**Keywords:** fruit flies, sexual selection, papaya fruit fly

## Abstract

**Simple Summary:**

Mating happens when males and females look for potential partners for reproduction. Each potential partner represents an array of qualities for each gender. Males may search for virgin, young, and large females as they offer many eggs for fertilization, while females may look for virgin, young, and large males as these may provide more food, sperm, and protection, or more extensive territories. However, nature can be counterintuitive, and the above statement is only sometimes true. Females prefer old males as mating partners, and females may not prefer virgin males for copulation. Identifying the attributes selected by each gender provides helpful information for implementing population management tactics and understanding the evolution of the species. *Anastrepha curvicauda* is a fruit fly that infests papaya fruit, rendering it unprofitable. Females do not discriminate between males, but males prefer to mate with virgin, young, and large females.

**Abstract:**

Sexual selection determines the evolution of the species by favoring some attributes that confer a reproductive advantage to those individuals with those attributes. Tephritidae flies do not always select the same traits when looking for a mating partner. Some aspects of the mating system of *Anastrepha curvicauda* are known; nevertheless, there is no information on the effect of age, size, and virginity when selecting a mating partner. We set up a series of experiments where a selector (male or female) may select between (a) an old or young partner, (b) a small or large partner, and (c) a virgin or mated partner. Males of *A. curvicauda* significantly preferred large, young, and virgin females, while females showed no preference for high- or low-quality males. The females’ non-preference for a particular male is discussed in the light of their mating system.

## 1. Introduction

Sexual selection determines the evolution of the species by selecting attributes that confer an advantage to specific individuals. These individuals are more likely to obtain a mating partner than those without those attributes [1,2,3]. Classic sexual selection theory argues that females are choosy because they invest more resources into reproduction than males, and males must experience the effects of sexual selection [4,5]. Therefore, virgin, young, and large/heavy individuals offer more benefits (in terms of resources and genes) than non-virgin, old, and small/light individuals [6,7], so a choosy young, virgin, and large/heavy female would be better off by selecting a virgin, young, and large/heavy male for mating [8].

Mate selection is a complex process that involves intrinsic (morphological, physiological, acoustic, tactile, or olfactive) and extrinsic (predations risk, operational sex ratio, environmental conditions, among others) factors and their interaction [8,9]. Female choosiness is related to the ability to distinguish the quality of potential mates during courtship and the potential benefits her choice provides [10,11,12,13].

Tephritidae sometimes choose different attributes when looking for potential mates. For example, *Ceratitis capitata* (Wiedemann) females prefer large [14] and young males for mating [15,16], but old males are accepted for mating when no young males are in the vicinity [17]. *Anastrepha obliqua* (Macquart) females prefer large males over small ones for mating regardless of whether they were obtained from a lab colony or from the field, but when large wild males compete with lab-reared large males, most matings are achieved by wild insects [18]. In *Anastrepha ludens* (Loew), females prefer old males for mating [2], but *Anastrepha suspensa* (Loew) females prefer young males [19]. In *Bactrocera tryoni* (Froggatt), young and large males achieve more matings than old and small individuals regardless of whether they are mated or virgin [20], and in *Zeugodacus cucurbitae* (Coquillett)*,* females do not discriminate between virgin and mated young males [21]. These examples show that females select partners depending on mate availability and their attributes across species.

Some Tephritidae flies such as *Anastrepha fraterculus* (Wiedemann), *C. capitata*, *Z. cucurbitae*, *A. ludens*, *A. obliqua*, and *Bactrocera* spp. form leks [21,22,23], where males agonistically compete for the best place inside the lek to display to females; mating occurs within the lek [24]. In these species, successful copulation depends more on displaying the proper behavior than on males fighting to access the females. Some others, such as *Rhagoletis indifferens* (Curran), *R. pomonella* (Walsh), and *Anastrepha curvicauda* Gerstaecker 1860 (former *Toxotrypana curvicauda*), do not form leks. Instead, males defend territories comprising plants of fruits from which they release a sexual pheromone to attract potential mates [24,25,26]. Access to females of these species depends on male attributes such as speed, maneuverability, fighting ability, or ability to defend a territory. These attributes are influenced by weight, age, and motivation for mating.

*Carica papaya* is the economically important host of *A. curvicauda* [27], and there are other hosts such as *Jacaratia mexicana* (Caricaceae) [28] and *Gonolobus sororius* (Asclepiadacea) [29]. *A. curvicauda* biology has been studied under field [25,27,30,31] and laboratory conditions [32,33,34,35,36,37]. Males arrive at papaya fruits before females and release their sexual pheromone (calling behavior) while perching on the fruit or leave. Up to four males have been recorded calling on the same papaya fruit [31] and chasing each other away. Most matings were observed at 13 h at Yautepec, Morelos [31]. Under laboratory conditions, newly emerged males may court females, which reach maturity at 6 d old [27] and may mate up to 4 times, while males mate up to 10 times [32], and both sexes live around 50 d [33].

Females do not need to feed on protein to produce eggs [27]. However, larval nutrition is paramount for their development and reproduction. *A. curvicauda* females obtained from mature *C. papaya* fruits [33] are heavier than those obtained from *J. mexicana* fruits [36], and females are heavier than males, but longevity on both hosts is similar for the two sexes [35,36]. Heavier females produce more corionated eggs than lighter ones, thus providing more reproductive benefits to their partner than the light ones [33,34]. *A. curvicauda* males establish a hierarchy where dominant males achieve most mating over subordinates [37], and mated males reduce their courtship vigor index [38].

Considering that *A. curvicauda* female and male attributes vary from individual to individual, these attributes change over time, and considering that virginity is a highly appreciated condition in a potential mate, we tested the influence of weight, age, and virginity on the insect mating process.

## 2. Materials and Methods

### 2.1. Insects

Mature larvae were obtained from infested papaya fruit at our experimental papaya grove at Yautepec, Morelos, Mexico [33]. Cylindrical plastic containers (500 mL capacity) with sterile soil (from our papaya grove) were used as pupation substrate. A total of 20 mL of water was added to the containers and covered with a cheesecloth secured by a rubber band. Cylindrical containers were kept at 50% relative humidity, 25 ± 2 °C, and a 12:12 h regime at our facilities.

Newly emerged flies (0 d) were weighed in an analytical Explorer Pro, Ohaus scale (Explorer, 0.0001 g accuracy, made in Switzerland) and contained individually in a 100 mL transparent cylindrical plastic container (3.6 cm diameter and 6.8 cm height covered with a piece of cheesecloth and a rubber band) to avoid any social interaction. Males and females were kept in separate rooms until needed. A ten percent sugar solution embedded in a cotton ball was provided as food. The average weight was 42.7 (ESM = 0.70) mg for males and 48.8 (ESM = 0.60) mg for females [33].

All experiments were performed at the Chemical Ecology lab under the above-mentioned environmental conditions between 11:30 and 16:00 h, the time of most fly activity at our location [30,31]. When two insects of the same sex were involved in an experiment, one was randomly marked on the thorax with a white dot using a white liquid Papermate pen corrector. This white dot did not restrain its movements nor its probability of achieving mating (binomial test, *p* > 0.05). Insects were used once and discarded.

### 2.2. Influence of Insect Weight on Mate Selection

This experiment tests the hypothesis that large/heavy individuals are preferred over small/light ones. Insects were classified as light, average (an average insect was defined as an insect whose weight falls between the mean ± ½ SD), or heavy, as shown in Table 1. For each replicate, we placed in the above-mentioned cylindrical container an average-weight insect (selector) with a light and a heavy insect (potential partners). We observed the triad for up to 2 h or until mating. All insects in this experiment were virgin and 6–8 d old. We performed 39 replicates for females and 30 for males as selectors.

### 2.3. Influence of Insect Age on Mate Selection

This experiment tests the hypothesis that young individuals are preferred over old ones. Virgin average-weight insects of different ages were used in this experiment. For each replicate, we used the same procedure of a selector (young) and two potential mates (young and old), as explained above. Individuals between 6 and 8 d old were classified as young, and those between 11 and 14 d as old. We tested 34 females and 30 males as selectors.

### 2.4. Influence of Mating Status on Mate Selection

This experiment tests the hypothesis that virgin individuals are preferred over mated ones. We used the same methodology as the two previous experiments except that the selectors were virgin individuals, and the selectees were virgin or mated (24 h in advance) insects. All insects in this experiment were 6–8 d old and of average weight. We tested 33 females and 30 males as selectors.

In all experiments, data on mate preference was analyzed by a binomial test (*p* = 0.05).

## 3. Results

### 3.1. Influence of Weight on Anastrepha curvicauda Mate Selection

Males of *A. curvicauda* significantly preferred heavy females over light ones for mating (*p* = 0.01); however, females did not discriminate among males (*p* = 0.09) (Table 2). The white dot on the thorax did not affect the probability of being selected (*p* = 0.08).

### 3.2. Influence of Age at Mating on Anastrepha curvicauda Mate Selection

Females did not discriminate between potential mates based on age (*p* > 0.05); however, males mated significantly more often with young females than with old ones (*p* < 0.03) (Table 3). The white dot on the thorax did not affect the probability of being selected (*p* = 0.09).

### 3.3. Effect of Mating Status on Anastrepha curvicauda Mate Choice

*Anastrepha curvicauda* females did not discriminate between potential mates based on mating status (*p* > 0.05); however, mated males chose virgin females significantly more often than mated (*p* < 0.001) (Table 4). The white dot on the thorax did not affect the probability of being selected (*p* = 0.08).

## 4. Discussion

Our results indicate that *A. curvicauda* males and females have different priorities when looking for a mate. Males selected virgin, large, and young females for mating, while females showed no preference for any of those attributes. Our findings counter the classical Darwinian idea that females should be choosy and males must experience the sexual selection process [5].

Female body weight indicates the number of eggs a female may produce or oviposit [33,39]. A heavy female has better quality and more eggs [39,40]. This is the case in *A. curvicauda*, where there is a linear and positive relationship between the female body weight of *A. curvicauda* and the number of corionated eggs ready for fertilization in females obtained from *C. papaya* [33] and *J. mexicana* [34]. This explains why *A. curvicauda* males selected heavy females for mating.

Many female fruit flies prefer large males for mating [40] as they may provide a better quality or larger ejaculate. For example, large and young *B. tryoni* females prefer large and young males as mating partners [20]. *Bactrocera oleae* (Rossi) females increase their fitness by identifying large good genetic quality males by the frequency of the vibration of their wing [41,42], and the wing vibration may be under sexual selection [4]. However, *A. ludens* and *A. fraterculus* females do not discriminate between large or small males for mating [18,43].

The wings of *A. curvicauda* and *A. suspensa* males are stubbier than those of the females and are morphologically modified to produce sound [41]. During courtship, *A. curvicauda, A. suspensa,* and *B. oleae* males produce sound by vibrating their wings, which correlates with male vigor [35,41,42]. Large *A. curvicauda* males (mean = 40 mg) obtain significantly more matings than small ones (mean = 22 mg) as larger males produce a higher-pressure level than small ones; unfortunately, no information on male age range or mating status was provided by the authors [35]. Our results do not support the latest statement, as females showed no mating preference based on body mass. This apparent contradiction deserves further investigation.

Male–male competition determines which males may court females. *A. curvicuada* males (6–8 d old and less than 0.05 mg bodyweight difference) establish a hierarchy by threatening, punching, pushing, or kicking [37]. Dominant males mate more often than subordinates. Our methodology did not allow for the establishment of a hierarchy among males, so its influence can be ruled out. Nevertheless, behavior plays an essential role in the reproductive success of *A. curvicauda,* as in *A. fraterculus* and *A. ludens* [10,18,43].

Age is important as old individuals offer fewer reproductive advantages than young ones, are less competitive, or lose their strength [44]. However, old males may be accepted as mating partners because they have lived enough to show their value as individuals (e.g., winning male–male fights or defending a territory), may transfer high-quality genes to their offspring [45], may be more competitive, or manage to produce a similar quality ejaculate as young ones [2].

Our results show that *A. curvicauda* females accepted young and old males as mating partners in similar proportions. In *C. capitata*, females prefer young and middle-aged males for mating because old males achieve 10% of mating when competing with young males [46]. *B. tryoni* females prefer young and large males for mating [20], and *A. suspensa* females also prefer young males as mating partners [19]. Conversely, wild females *A. ludens* preferred older and mated males for mating regardless, females did not increase their reproductive success in terms of offspring or longevity [2].

Mate selection is critical for females with a limited opportunity to find sexual partners and obtain an ejaculate; the relevance of the selection diminishes when sexually mature individuals are available throughout the mating season. Sperm seems not to be a limiting resource for *A. curvicauda* females as they may mate up to four times and males ten times in their lifetime [32]. This secures an almost unlimited supply of sperm and male accessory gland secretions. Whether multiple mating increases fecundity or longevity in *A. curvicauda* females remains unknown.

*Anastrepha curvicauda* males may assess their potential partner’s age and sexual maturity through behavioral cues during courtship or by the composition of their cuticular hydrocarbon profiles. Males touch females with their forelegs during courtship before mounting them and attempting mating [27,37]. Females correspond by touching males before allowing mounting by males. The compound 1-heptacosanol is the main compound in *A. curvicuada* females, and an increase is observed when they are sexually mature (7 d old) [47]. A similar situation has been reported for *A. obliqua*, where the cuticular hydrocarbon n-nanocosene correlates with old individuals of both genders, and higher levels of n-heneicosene, n-heneicosane, and n-tricosene differentiate males from females [48]. In *A. fraterculus,* mature females and males have specific cuticular hydrocarbon profiles for each gender [49].

Contrary to *A. curvicauda* females, young and old *D. melanogaster* females are equally attractive to males for mating [50]. *C. capitata* young and old females do not discriminate among males on age when their age difference is less than 10 d but do against 40 d old males in the presence of 10 d old ones. Nevertheless, mating with a young male does not provide any advantage regarding fecundity and fertility. Males with a higher mating rate are preferred for mating [21,51], so sperm is not a limited factor for *C. capitata* females.

A meta-analysis [7] revealed that male quality might vary, but all males preferred good-quality females (regarding reproductive benefits for males) for mating. In our research, males were large or small, old or young, and mated or virgin (good- or low-quality male), and all of them preferred large, young, and virgin females for mating. In this sense, virginity, body size, and age are desirable attributes for males when selecting a mating partner. Regardless of the fact that the meta-analysis [7] excluded choice experiments from their analysis, their results align with ours.

Thus, why do *A. curvicauda* females accept mating with non-virgin, old, and small males? The reasons could be related to their mating system. Males defend papaya fruits from other males where they release their sexual pheromone [30,52], and females are lured on to the fruits for mating [52]; the fruits represent oviposition sites where females lay their eggs [31,52]. A papaya tree may host 3+ males looking for mating partners [31]. These males represent an almost unlimited supply of sperm and accessory glands secretions. Females may have up to four matings and males up to ten under lab conditions [32]. In a 10 d field cage experiment where ten females and ten males were released, three acts of courting and three matings were observed on the papaya fruit [53]. It is possible that more matings would have been observed if the experiment had lasted longer.

In our experience working on papaya plantations, papaya plants have many fruits, most of which are populated by one or two males in addition to those found in the leaves. Additionally, on several occasions, we have observed males attempting to mate, the females rejecting the attempt by kicking the male’s body with their hindlegs, and males forcing the ovipositor into copulation or a male on top of a couple in mating. The two previously mentioned behaviors were not observed under our experimental conditions, but they were observed in the lab cages, and the tandem behavior [37] was observed using the same container we used in our experiments. Using a larger container could facilitate the appearance of behaviors other than those observed, influencing the results obtained in our research. However, intrasexual female competition is more about resources and reproductive opportunities. It is about ovipositing sites, social status, breeding places, access to food, etc., and not for mating partners. The intrasexual competition among females may create individual differences: larger females with more eggs are preferred by the males [54].

Several theories have been proposed as explanations for the speciation process and genetic variation [55]. All predict that the continuous selection of a specific trait by any sex due to sexual selection may cause slight changes in time that would cause reproductive isolation. For example, early speciation of *A. fraterculus* has been identified by the analysis of the calling behavior pattern, differences in the pheromone time release, and the reproductive performance of the offspring, among others [55]. Our paper does not report any genetic study but reports on traits (size, age, mating status) that may be under sexual selection. How female sexual selection operates, how sexual differences evolved, and what they compete for remain to be discovered [56].

## 5. Conclusions

Under our experimental conditions, *A. curvicauda* males preferred virgin, young, and heavy females for mating over non-virgin, old, and light females. Unlike males, females did not discriminate between non-virgin and virgin, young and old, and light or heavy males. This apparent non-selection by females may be explained partially by a constant supply of sperm during the mating season and the fact that females may copulate several times in their lifetime. This is the first study on the effect of these attributes on the mate selection process in *A. curvicauda*.

## Figures and Tables

**Table 1 insects-14-00317-t001:** Classification of *Anastrepha curvicauda* females and males according to their weight.

Classification	Females’ Weight (mg)	Males’ Weight (mg)
Light	15 to 35	13 to 36
Average	36 to 50	37 to 48
Heavy	51 to 89	49 to 75

**Table 2 insects-14-00317-t002:** Influence of body weight on *A. curvicauda* mate selection. All insects were 6–8 d.

Selectors		Selectees	
	Light	Heavy	n
Female	17	22	39
Male	9	21 *	30

* *p* = 0.01.

**Table 3 insects-14-00317-t003:** Influence of age at mating on *A. curvicauda* on mate selection.

Selectors	Selectees		
	Young (6–8 d old)	Old (11–13 d old)	n
Female	15	15	30
Male	22 *	12	34

* *p* = 0.03.

**Table 4 insects-14-00317-t004:** Influence of mating status on *A. curvicauda* mate selection. All insects were 6–8 d.

Selectors	Selectees		
	Virgin	Mate	n
Female	17	16	33
Male	24 *	6	30

* *p* < 0.001.

## Data Availability

All data are presented in the text.

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
