# Peer review of "Mate Selection on Anastrepha curvicauda: Effect of Weight, Age, and Virginity"

_insects, 2023, doi:10.3390/insects14040317_

Round 1
Reviewer 1 Report
This manuscript is very well written and contains makes an important contribution to the field of mating behavior in tephritids. My comments are only minor editorial suggestions.
Line 82 should be ‘species’
Line 146-147 be consistent with spacing after and before < or = signs
Caption for tables 2 and 4 should probably have the age of the flies tested there.
Line 165 extra period
Author Response
Reviewer 1. Why Anastrepha curvicuada females are not choosy?
Line 82 should be ‘species’
REPLY: Done accordingly. See line 85.
Line 146-147 be consistent with spacing after and before < or = signs
REPLY: All the spaces have been reviewed.
Caption for tables 2 and 4 should probably have the age of the flies tested there.
REPLY: The following text has been amended to the title of Tables 2 and 4. “All insects were 6-8 d.”
Line 165 extra period
REPLY: Done accordingly. See line 167.
Thanks for your valuable comments.
Reviewer 2 Report
This manuscript describes some simple mating experiments in the laboratory with a fruit fly species. The authors are attempting to draw conclusions about sexual selection in their flies, using flies enclosed in small plastic vials, whereas the natural arenas for mating in this species are papaya fruits where males establish and defend territories, and attract females. Because these conditions are so different than the inside of a small featureless plastic vial, it is unclear how well the results and conclusions may reflect the realities of this insect’s mating system. This crucially important point has not really been addressed anywhere in the manuscript, but should be.
Specific comments as follows:
1. The authors have phrased their title as a question, i.e., why are females not choosy, but they have not really answered this in the text. They need to do a much better job of offering explanations for why females did not discriminate between young/old or large/small males. Does this reflect reality and if so why, or might it simply be an artefact of the very artificial bioassay conditions, where males may not have had a chance to sing, display, call, or whatever it is that they do during courtship under natural conditions. That is, on L 48-50, the authors have described the complex environment under which mating takes place under natural conditions, which is not at all duplicated by a small closed plastic vial. This must be discussed
2. L 16: species not specie, and infest, not larvae
3. L 33, meaning of apport unclear, clarify
4. L 39, define OSR
5. L 80, how can knowledge of the mating system be used in management programs? Either explicitly describe how this knowledge can be exploited, or delete this statement.
6. L 81-83, this is general and vague. If it is true, then explicitly state how a more detailed knowledge of the mating system may shed light on genetic variation and speciation. Otherwise, these are just buzzwords dropped in to spice up the paper, with no real meaning.
7. L 87, delete first sentence, which is repeated in the second sentence. Also, provide more detail, i.e., were mature larvae simply dropped into the containers singly, or in groups? Was food added?
8. L 98m what is ESM?
9. L 122, earlier the authors had stated that flies live up to 50 days or so. Here, their old fly cohort consists of flies 11-14 d old. Why did they use relatively young flies still, when they can live much longer?
10. In the discussion, it might be useful to discuss how flies are able to assess the age and mating status of each other. Is this mediated by chemical cues, or physical cues, or…? Are there any clues in the behavior, i.e., to males antennate females before trying to mate, and do females antennate males as well?
11. L 160-162, as mentioned above, what are the possible explanations for females not appearing to be choosy? Is this a real biological phenomenon, or is it possible or likely to be an artefact of the very unnatural mating conditions, with flies confined in small containers instead of being on papaya fruit? If the authors had done their bioassays by putting flies on papaya fruit, is it likely that they would have gotten different results?
12. L 167, fertilization, not fecundation
13. L 169-172, so what signals or cues do female A curvicauda females use to recognize and assess males? I.e., presumably not all mating attempts by males are successful, so females must be using some cues to reject males some of the time. In fact, given that the authors found no evidence for female choice in their bioassays, it would be very useful and enlightening to know what percentage of mating attempts by males are successful. If mating attempts by males are always successful, that would argue for no female choice. However, if a significant proportion of mating attempts are unsuccessful, that would argue that females are indeed being choosy. Comments please.
14. L 169-172: is there any information on the size and quality of the ejaculate, i.e., is it small and consisting primarily of sperm, or is it large, consisting of both sperm and nutrients to help with development of more eggs? Again, this information is a crucial piece of the puzzle in terms of sexual selection, in terms of costs to males and benefits to females.
15. L 182-183, the fact that your results do not agree with previous results again suggests that your results in terms of no female choice of large or small males may be an experimental artefact. This is further suggested by L 184-189, where you state that the typical dominance hierarchies were not established under your bioassay conditions, and if the dominance hierarchies are a crucial part of the mating behavior, then in your bioassays, females might have lacked crucial information for making a choice. Comments?
16. L 192-204 lays out different scenarios, i.e., in some cases, old males are better, whereas in others, they are worse. However, you did not follow up to show how your own system fits into one or the other of these patterns, or possibly forms its own pattern. That is, besides describing what happens in a number of different systems, describe how some of these results relate directly to your system.
17. L 207-211: you state that females mate up to 4 times, and males up to 10 times. However, I presume that these results were obtained under lab conditions. In the field, I suspect that things are much different, both because mates would be much more scarce than in small closed containers, and because most flies would probably be killed by predation after a few days. Thus, it is very unlikely under field conditions that females would experience an unlimited supply of sperm and secretions as suggested here and on L 240. This should be rewritten in terms of the natural biology, rather than lab results obtained under very artificial conditions.
18. L 212-219: as written, this proposed use of cuticular lipids to assess age and mating status is completely speculative. Can the authors provide some support for this by describing in detail the close range mating behaviors. That is, is there antennation or palpation with mouthparts, when cuticular chemicals could be assessed? IS there any other evidence that directly or indirectly would implicate the use of CHCs in mate choice in this species?
19. L 216-217, the names of the cuticular hydrocarbons have been mangled.
20. L 235-250 again demonstrates how the lab bioassay conditions may have distorted the results. As stated, in the small containers, the males would not be able to set up territories to defend and lure females because there was no fruit present, nor could they set up dominance hierarchies. Second, if they released pheromone, inside these small closed containers, the females could probably not tell which male was producing pheromone. Thus, the females may have had few or none of the normal signals and cues that would allow them to choose among available males.
21. It would have been useful and enlightening if the authors had determined the number of eggs laid and the hatch rates. That is, did matings with old, already mated, and small males result in a fitness cost to females? If it did not, then this would provide support for females not being choosy. However, if there WAS a fitness cost, then this would suggest that the bioassay used was not adequate to assess female choice properly. Comments?
Reviewer 3 Report
Manuscript ID: insects-2219950
Type of manuscript: Article
Title: Why are Anastrepha curvicauda females not choosy?
In this study titled " Why are Anastrepha curvicauda females not choosy?”, mate choice by mate conditions (age and body size) and mating status of mates were investigated in Anastrepha curvicauda. From choice test, it is founded that males preferred high-quality (young, large and virgin) females, while females discriminated male quality as mate. Experiments and the results are very clear and these results are very interesting. However, it needs to revise the frame of Introduction and Discussion. Comments are as below.
Major comments
In introduction (L43-60), I think that authors want to claim that traits preferred by mate (male and female) are different by mating system, environment situation and individual condition at mating event between related species and even in the species. However, these paragraphs were not clear the stream of purpose of this study. These paragraphs imply that aim of this study is comparison of mating system, related species or status of individuals on mate choice. It is needed to rewrite the framework to show straightforward.
As same with Introduction, Discussion was not clear what did results mean in sexual selection (mate choice) studies? These results were compared with other species but most of these comparison didn’t discuss why different or same with other species from perspective of sexual selection (L192-226).
I didn’t understand the state of last paragraph. Why the authors concluded that A. curvicauda females ARE LOOKING FOR something other than good-quality males? Females have no mate preference by their condition and mating status. This didn’t mean female are looking for other.
As authors mentioned in this manuscript, male competes over territory (food and oviposit resource) and releases pheromone to attract females. Male would have risk to decrease his fitness by allowing to mating with non-virgin females and their oviposition because it is possible that the females oviposit offspring sired previous mates. On the other hands, females attracted to a male with territory would not need to assess his quality because he is superior to in male-male competition, i.e., high quality male. Together with mating system of this species, results of this study should be discussed.
In this study, results (no. in tables) show no. of mating pairs (right?). From these results, it was concluded that males preferred light, young and mated females. However, it is possible that heavy, old and mated female rejected the mate. Do you have any data which show that heavy, old and mated female didn’t reject and male preferred light, young and mated females?
L169: Avoid to use many or define the area of “many”. Many females among taxa? Many females of this species?
Remove L13 (Some females …). This sentence would suggest variations of mate preference in the species or population, but this is out of your results and the readers would confuse.
L24: What means high-quality?
Minor
L33: apport -> apportion? What means “resources”? Gamete size? Does it mean cost of multiple mating?
L53: Add “.” After [24] .
L68: lab -> laboratory
L68-70: avoid to use ‘may’. These are results of previous studies.
L164: Remove “ … and may build large population quickly.” This is unnecessary in this paragraph.
L188-189: Add the sentence as a note in materials and method.
Round 2
Reviewer 2 Report
The authors have improved the manuscript. However, they have not addressed two major points:
1. the justification for the studies, in terms of providing information that could be exploited for insect control, and for speciation and genetics studies.
2, They have not discussed at all whether their results truly reflect what happens in the field, or whether at least some of their results may be wholly or in part, artefacts of the very artificial conditions of their laboratory bioassays. It is CRUCIALLY important to clearly, explicitly, and thoughtfully discuss this.
Specific comments to their rebuttals below:
L 80, how can knowledge of the mating system be used in management programs? Either explicitly describe how this knowledge can be exploited, or delete this statement.
Reply: Establishing the highly successful fruit fly programs against the medfly (Ceratitis capitata)or the mexfly (Anastrepha ludens)required detailed knowledge of their mating system. The selection of individuals for the rearing stock required many details of the relationship between food supply-bodyweight-fecundity, oviposition curves to secure the most viable eggs, the response of mate and virgin females to the sex pheromone or
the effect of male-male competition on mate choice. We could include some of this information in the paper, but we consider it may mislead the reader’s attention as our results are quite specific.
è I disagree. One of the major justifications that the authors have stated for this work is to provide detailed information about the mating systems that might be exploited for management of the fly. Thus, having stated this, they should describe explicitly how the work described here will address that justification. Otherwise, as stated, the statement is just a vague sales pitch.
6.L 81-83, this is general and vague. If it is true, then explicitly state how a moredetailed knowledge of the mating system may shed light on genetic variation and speciation. Otherwise, these are just buzzwords dropped in to spice up the paper, with no real meaning.
REPLY: Several theories propose an explanation for the speciation process and genetic variation. All agree that the continuous selection of a specific trait by any gender due to sexual selection may cause slight changes in time, either in the selector or the selectee, that would render reproductive isolation Our paper does not report any genetic study but does report on traits (size, age, mating status) that may be under sexual selection.
è So this explanation should be included in the text. Otherwise, as stated, it is just more general and vague sales pitch. Tell the reader how and why it could be important.
11. L 160-162, as mentioned above, what are the possible explanations for females not appearing to be choosy? Is this a real biological phenomenon, or is it possible or likely to be an artefact of the very unnatural mating conditions, with flies confined in small containers instead of being on papaya fruit? If the authors had done their bioassays by putting flies on papaya fruit, is it likely that they would have gotten different results?
REPLY: The study of the biology of any insect under lab conditions is always controversial. We acknowledge that a lab environment is an abstraction of the natural environment. If more stimuli are involved in the mating selection process, like host odor, conspecifics, pheromone release, sound, etc., it is likely that the results would vary from what we report. However, in this case, males do not need to release a pheromone to call a female as the female is in the container, and host odor is not mandatory to display courtship and achieve mating as in other fruit flies. A. curvicauda males produce sound during courtship (Sivinski and Weeb 1985, reference no. 46 of the manuscript), and the container does not prevent them from producing sound. We consider that the numerical results may differ, but the phenomenon would remain.
è The authors have not addressed the point made, i.e., they have not provided any additional thoughtful discussion about how their results may be a result of the bioassay conditions rather than reflecting the natural conditions. It is crucially important to include this in the discussion because otherwise, it could propagate a falsehood in the literature, i.e., that females are not choosy, whereas in reality, in the field females could very easily be, and even are likely to be choosy. Also, the authors state that the male do not need to release a pheromone to attract females because they are already in close proximity to the females in the small vial. However, this ignores the fact that pheromone release could be a key component of courtship. In addition, the authors state that the small container does not prevent the males from producing sound. However, much more to the point is not whether they can, but whether they will, i.e., do they still produce courtship sound under these very artificial conditions? If they do, this should be stated explicitly.
13. L 169-172, so what signals or cues do female A curvicauda females use to recognize and assess males? I.e., presumably not all mating attempts by males are successful, so females must be using some cues to reject males some of the time. In fact, given that the authors found no evidence for female choice in their bioassays, it would be very useful and enlightening to know what percentage of mating attempts by males are successful. If
mating attempts by males are always successful, that would argue for no female choice. However, if a significant proportion of mating attempts are unsuccessful, that would argue that females are indeed being choosy. Comments please.
REPLY: Flies may use the different profiles of cuticular hydrocarbons (CHC) of individuals of different gender and age. Lines 211 to 214 comment on a previous paper by a former student of our group. CHC profiles of virgin, mated, young and old individuals are different. We are working on bioassays to show the recognition of the CHC by males and females.
Unfortunately, we did not quantify the number of unsuccessfully mating attempts by males but mating propensity is over 80%, as mentioned in reply to question no. 9.
è So in the discussion, the authors might want to explicitly mention that because not all male mating attempts are successful, females must be using some kind of criteria for rejecting at least some males, and so they are at least slightly choosy? Or is the rejection of some males possibly due to the females not being physiologically ready to mate or something like that, i.e., that for those females, no male, no matter how high quality, would be acceptable?
14. L 169-172: is there any information on the size and quality of the ejaculate, i.e., is it small and consisting primarily of sperm, or is it large, consisting of both sperm and nutrients to help with development of more eggs? Again, this information is a crucial piece of the puzzle in terms of sexual selection, in terms of costs to males and benefits to females.
REPLY: To the best of our knowledge, there is no information on ejaculate size, number of sperms, or accessory glands secretions per ejaculate. We unsuccessfully tried to quantify sperm.
è So have you tried weighing both males and females before and after mating, to see if there is a significant weight change? Just a comment, but could be informative to try. I.e, if you can document a significant weight loss by males and a significant weight gain by females, that could give you some idea of the male investment.
Author Response
- the justification for the studies, in terms of providing information that could be exploited for insect control, and for speciation and genetics studies.
REPLY: The following text has been included in the Discussion section of the manuscript as indicated by the reviewer “Several theories propose an explanation for the speciation process and genetic variation. All agree that the continuous selection of a specific trait by any gender due to sexual selection may cause slight changes in time, either in the selector or the selectee, that would render reproductive isolation. For example, early speciation of A. fraterculus has been identified by the analysis of the calling behavior pattern, differences in the pheromone time release, and the reproductive performance of the offspring, among others”. See lines 277-281.
2, They have not discussed at all whether their results truly reflect what happens in the field, or whether at least some of their results may be wholly or in part, artefacts of the very artificial conditions of their laboratory bioassays. It is CRUCIALLY important to clearly, explicitly, and thoughtfully discuss this.
REPLY: The following text has been included in the manuscript. “Also, on several occasions, we have observed males attempting to mate, the females rejecting the attempt by kicking the male’s body with their hindlegs, and males forcing the ovipositor into copulation or a male on top of a couple in mating. The two previously mentioned behavior were not observed under our experimental conditions, but they were observed in the lab cages, and the tandem behavior [39] was observed using the same container we used in our experiments. Using a larger container could facilitate the appearance of behaviors other than those observed, influencing the results obtained in our research”. See lines 275-281.
Specific comments to their rebuttals below:
L 80, how can knowledge of the mating system be used in management programs? Either explicitly describe how this knowledge can be exploited, or delete this statement.
Reply: Establishing the highly successful fruit fly programs against the medfly (Ceratitis capitata)or the mexfly (Anastrepha ludens) required detailed knowledge of their mating system. The selection of individuals for the rearing stock required many details of the relationship between food supply-bodyweight-fecundity, oviposition curves to secure the most viable eggs, the response of mate and virgin females to the sex pheromone or the effect of male-male competition on mate choice. We could include some of this information in the paper, but we consider it may mislead the reader’s attention as our results are quite specific.
I disagree. One of the major justifications that the authors have stated for this work is to provide detailed information about the mating systems that might be exploited for management of the fly. Thus, having stated this, they should describe explicitly how the work described here will address that justification. Otherwise, as stated, the statement is just a vague sales pitch.
REPLY. We have deleted the mentioned statement. See lines 105-108
6.L 81-83, this is general and vague. If it is true, then explicitly state how a more detailed knowledge of the mating system may shed light on genetic variation and speciation. Otherwise, these are just buzzwords dropped in to spice up the paper, with no real meaning.
REPLY: Several theories propose an explanation for the speciation process and genetic variation. All agree that the continuous selection of a specific trait by any gender due to sexual selection may cause slight changes in time, either in the selector or the selectee, that would render reproductive isolation Our paper does not report any genetic study but does report on traits (size, age, mating status) that may be under sexual selection.
So this explanation should be included in the text. Otherwise, as stated, it is just more general and vague sales pitch. Tell the reader how and why it could be important.
REPLY: We included this information in the text. See lines 277-281.
- L 160-162, as mentioned above, what are the possible explanations for females not appearing to be choosy? Is this a real biological phenomenon, or is it possible or likely to be an artefact of the very unnatural mating conditions, with flies confined in small containers instead of being on papaya fruit? If the authors had done their bioassays by putting flies on papaya fruit, is it likely that they would have gotten different results?
REPLY: The study of the biology of any insect under lab conditions is always controversial. We acknowledge that a lab environment is an abstraction of the natural environment. If more stimuli are involved in the mating selection process, like host odor, conspecifics, pheromone release, sound, etc., it is likely that the results would vary from what we report. However, in this case, males do not need to release a pheromone to call a female as the female is in the container, and host odor is not mandatory to display courtship and achieve mating as in other fruit flies. A. curvicauda males produce sound during courtship (Sivinski and Weeb 1985, reference no. 46 of the manuscript), and the container does not prevent them from producing sound. We consider that the numerical results may differ, but the phenomenon would remain.
The authors have not addressed the point made, i.e., they have not provided any additional thoughtful discussion about how their results may be a result of the bioassay conditions rather than reflecting the natural conditions. It is crucially important to include this in the discussion because otherwise, it could propagate a falsehood in the literature, i.e., that females are not choosy, whereas in reality, in the field females could very easily be, and even are likely to be choosy. REPLY: Please see lines 275-281. Also, the authors state that the male do not need to release a pheromone to attract females because they are already in close proximity to the females in the small vial. However, this ignores the fact that pheromone release could be a key component of courtship. REPLY: to the best of our knowledge, the A. curvicuada pheromone is a long-distance pheromone. We have observed mating couples under lab conditions where the males do not expand their pouches prior to mating. In addition, the authors state that the small container does not prevent the males from producing sound. However, much more to the point is not whether they can, but whether they will, i.e., do they still produce courtship sound under these very artificial conditions? If they do, this should be stated explicitly. REPLY: We consider that the fact that matings were observed in our small containers indicates that even males produce sound, or it is not necessary.
We decided to change the title of the paper to “Mate selection on Anastrepha curvicauda: effect of weight, age and virginity”
- L 169-172,so what signals or cues do female A curvicauda females use to recognize and assess males? I.e., presumably not all mating attempts by males are successful, so females must be using some cues to reject males some of the time. In fact, given that the authors found no evidence for female choice in their bioassays, it would be very useful and enlightening to know what percentage of mating attempts by males are successful. If mating attempts by males are always successful, that would argue for no female choice. However, if a significant proportion of mating attempts are unsuccessful, that would argue that females are indeed being choosy. Comments please
REPLY: Flies may use the different profiles of cuticular hydrocarbons (CHC) of individuals of different gender and age. Lines 211 to 214 comment on a previous paper by a former student of our group. CHC profiles of virgin, mated, young and old individuals are different. We are working on bioassays to show the recognition of the CHC by males and females. Unfortunately, we did not quantify the number of unsuccessfully mating attempts by males but mating propensity is over 80%, as mentioned in reply to question no. 9.
So in the discussion, the authors might want to explicitly mention that because not all male mating attempts are successful, females must be using some kind of criteria for rejecting at least some males, and so they are at least slightly choosy? Or is the rejection of some males possibly due to the females not being physiologically ready to mate or something like that, i.e., that for those females, no male, no matter how high quality, would be acceptable?
REPLY. All females used in the experiments were sexually mature, as mentioned in lines 89-90 of the text. See lines 275-281
- L 169-172: is there any information on the size and quality of the ejaculate, i.e., is it small and consisting primarily of sperm, or is it large, consisting of both sperm and nutrients to help with development of more eggs? Again, this information is a crucial piece of the puzzle in terms of sexual selection, in terms of costs to males and benefits to females.
- REPLY: To the best of our knowledge, there is no information on ejaculate size, number of sperms, or accessory glands secretions per ejaculate. We unsuccessfully tried to quantify sperm
So have you tried weighing both males and females before and after mating, to see if there is a significant weight change? Just a comment, but could be informative to try. I.e, if you can document a significant weight loss by males and a significant weight gain by females, that could give you some idea of the male investment.
REPLY: We weighed the insects before and after mating and failed to detect any weight difference. We used the Analytical Explorer Pro, Ohaus scale (Explorer, 0.0001 g accuracy made in Switzerland) reported in lines 118-119.
Reviewer 3 Report
Manuscript ID: insects-2219950
Type of manuscript: Article
Title: Why are Anastrepha curvicauda females not choosy?
In previous comments, I commented that
“In introduction (L43-60), I think that authors want to claim that traits preferred by mate (male and female) are different by mating system, environment situation and individual condition at mating event between related species and even in the species. However, these paragraphs were not clear the stream of purpose of this study. These paragraphs imply that aim of this study is comparison of mating system, related species or status of individuals on mate choice. It is needed to rewrite the framework to show straightforward.”. And, the authors replied that
“We apologize for the inconvenience. The following sentence has been amended to the text “The objective of our paper was to address the role that weight, age, or mating status play in this insect’s mate choice” See line 81-82. “ However, this reply didn’t response my comment correctly.
I believe that the authors should clarify the purpose of including paragraphs 3 to 6 in the Introduction section. It seems that authors want to claim that Tephiritidae flies have various mating systems (Paragraph3) and mating choice plasticity (paragraph 4) depending on species. In Anastrepha curvicauda has several hosts and the host difference cause differences of individual condition and fitness (i.e., body weight and fecundity) (Paragraph 5 and 6). So, The authors aim to investigate the potential effects of individual condition and mating status on mate choice (paragraph 7). However, the connection between these paragraphs and the overall objective of the paper is not clear. Merely adding a sentence such as 'The objective of our paper was to address the role that weight, age, or mating status play in this insect’s mate choice' is insufficient to fully convey the study's purpose and rationale.
Author Response
In previous comments, I commented that
“In introduction (L43-60), I think that authors want to claim that traits preferred by mate (male and female) are different by mating system, environment situation and individual condition at mating event between related species and even in the species. However, these paragraphs were not clear the stream of purpose of this study. These paragraphs imply that aim of this study is comparison of mating system, related species or status of individuals on mate choice. It is needed to rewrite the framework to show straightforward.”. And, the authors replied that
“We apologize for the inconvenience. The following sentence has been amended to the text “The objective of our paper was to address the role that weight, age, or mating status play in this insect’s mate choice” See line 81-82. “ However, this reply didn’t response my comment correctly.
I believe that the authors should clarify the purpose of including paragraphs 3 to 6 in the Introduction section. It seems that authors want to claim that Tephiritidae flies have various mating systems (Paragraph3) and mating choice plasticity (paragraph 4) depending on species. In Anastrepha curvicauda has several hosts and the host difference cause differences of individual condition and fitness (i.e., body weight and fecundity) (Paragraph 5 and 6). So, The authors aim to investigate the potential effects of individual condition and mating status on mate choice (paragraph 7). However, the connection between these paragraphs and the overall objective of the paper is not clear. Merely adding a sentence such as 'The objective of our paper was to address the role that weight, age, or mating status play in this insect’s mate choice' is insufficient to fully convey the study's purpose and rationale.
REPLY: We restructured the introduction to focus more on the individual attributes than on the Tephritidae mating systems. Also, the text includes a small explanation of each mating system's main characteristics. Former paragraph number three is now number four. See lines 49 to 81 and 101 to 104.
Round 3
Reviewer 2 Report
I thank the authors for making a further round of revisions. If they are happy with the revised manuscript, I have nothing further to say
Author Response
We thank the reviewer for their useful comments